

# Detection of methylation, acetylation and glycosylation of protein residues by monitoring $^{13}C$ chemical-shift changes: A quantum-chemical study

Pablo G. Garay[1], Osvaldo A. Martin[1], Harold A. Scheraga[2] and Jorge A. Vila[1]

[1] IMASL-CONICET, Universidad Nacional de San Luis, San Luis, Argentina
[2] Chemistry and Chemical Biology, Cornell University, Ithaca, NY, USA

## ABSTRACT

Post-translational modifications of proteins expand the diversity of the proteome by several orders of magnitude and have a profound effect on several biological processes. Their detection by experimental methods is not free of limitations such as the amount of sample needed or the use of destructive procedures to obtain the sample. Certainly, new approaches are needed and, therefore, we explore here the feasibility of using $^{13}C$ chemical shifts of different nuclei to detect methylation, acetylation and glycosylation of protein residues by monitoring the deviation of the $^{13}C$ chemical shifts from the expected (mean) experimental value of the non-modified residue. As a proof-of-concept, we used $^{13}C$ chemical shifts, computed at the DFT-level of theory, to test this hypothesis. Moreover, as a validation test of this approach, we compare our theoretical computations of the $^{13}C_\varepsilon$ chemical-shift values against existing experimental data, obtained from NMR spectroscopy, for methylated and acetylated lysine residues with good agreement within ∼1 ppm. Then, further use of this approach to select the most suitable $^{13}C$-nucleus, with which to determine other modifications commonly seen, such as methylation of arginine and glycosylation of serine, asparagine and threonine, shows encouraging results.

## INTRODUCTION

Since the pioneer observation of lysine methylation of a bacterial protein by *Ambler & Rees (1959)*, there has been a rising interest in investigating protein post-translational modifications (PTMs) and their role as modulators of protein activity (*Zobel-Thropp, Gary & Clarke, 1998*; *Bienkiewicz & Lumb, 1999*; *Bannister, Schneider & Kouzarides, 2002*; *Paik, Paik & Kim, 2007*; *Bedford & Clarke, 2009*; *Kamieniarz & Schneider, 2009*; *Kamath, Vasavada & Srivastava, 2011*; *Luo, 2012*; *Theillet et al., 2012a*; *Theillet et al., 2012b*; *Evich et al., 2015*; *Rahimi & Costello, 2015*; *Schubert et al., 2015*). As a consequence of their relevance, the development of fast and accurate experimental methods for detection of PTMs has also been an object of active research in the field (*Kamath, Vasavada & Srivastava, 2011*). In this regard, *Schubert et al. (2015)* recently proposed a novel NMR-based

Corresponding author
Jorge A. Vila, jv84@cornell.edu

methodology to detect glycosylation of residues in proteins under urea-denaturing conditions. The advantages and disadvantages of the proposed new methodology against existing methods for the analysis and detection of PTMs, such as Mass Spectroscopy (MS) (*Kamath, Vasavada & Srivastava, 2011*; *Luo, 2012*), were discussed in detail by *Schubert et al. (2015)*. One of the main disadvantages of this new methodology (*Schubert et al., 2015*) is the several larger orders of magnitude of sample required, compared to that needed in MS experiments. Despite this limitation, and the requirement of urea-denaturing conditions, the methodology of *Schubert et al. (2015)* presents interesting advantages over existing methods, such as analysis of intact proteins to allow the detection of glycosylation in proteins as well as to identify the composition of the attached glycans and the type of glycan linkages. The use of chemical-shift variations to sense PTMs is not a novel approach (*Bienkiewicz & Lumb, 1999*). In fact, it was used to analyze random-coil chemical-shift variations of the $^{1}$H, $^{13}$C and $^{15}$N nuclei of serine, threonine and tyrosine upon phosphorylation (*Bienkiewicz & Lumb, 1999*). The results were very promising, e.g., up to ∼4 ppm chemical-shift difference was observed for the $^{13}$C$^{\beta}$ nucleus of Thr upon phosphorylation. However, to extend this approach to treat other PTMs would require the monitoring of chemical-shift changes in random-coil model peptides of these PTMs, and this is a costly and time consuming procedure. For this reason, we propose such monitoring *in silico,* rather than by using random-coil experiments, as a method to identify the most suitable $^{13}$C nuclei with which to sense the existence of PTMs in proteins, e.g., to detect the states of arginine methylation, or glycosylation of serine, asparagine and threonine residues.

We propose to analyze the feasibility to detect PTMs by measuring the deviation of the $^{13}$C chemical shifts of a given nucleus from its mean experimental value. To evaluate this idea, in silico, we proceed as follows. For a selected nucleus, we compute, at the DFT-level of theory for an ensemble of conformations, the $^{13}$C chemical-shifts for the *non*-modified and the modified residues, respectively. We built these ensembles as a set of pairs of modified and *non*-modified residues possessing the same backbone and side-chain torsional angles. Consequently, the only difference in the computed chemical shifts between both distributions came from the PTM. In fact, although chemical shifts are influenced by both the torsional angles and the PTM, the former effect is canceled out because both distributions possess the same set of torsional angles. Then, from each member of both ensembles we subtract the mean of the $^{13}$C chemical-shift of the ensemble from the *non*-modified residue. This is what we call, from here on, the Δ value. As a result, the computed Δ values are the expected deviations from the *non*-modified mean chemical shift values. Naturally, the distribution of the Δ values for *non*-modified residues is, by definition, centered around 0.0 ppm, as can be seen for each of the blue-line curves in Figs. 1–7 (except Figs. 3 and 4).

The use of Δ values to detect PTMs does not require carrying out experiments under urea-denaturing conditions or the use of a large amount of sample, as with the recent NMR-based proposed methodology (*Schubert et al., 2015*). This is an advantage. As a disadvantage, it does require the labeling of the carbons of the protein, although $^{13}$C-labeling is a widely and standard procedure used in both NMR-based experiments *and* theoretical studies because of the exquisite sensitivity of the chemical-shifts to: (i) identify flaws

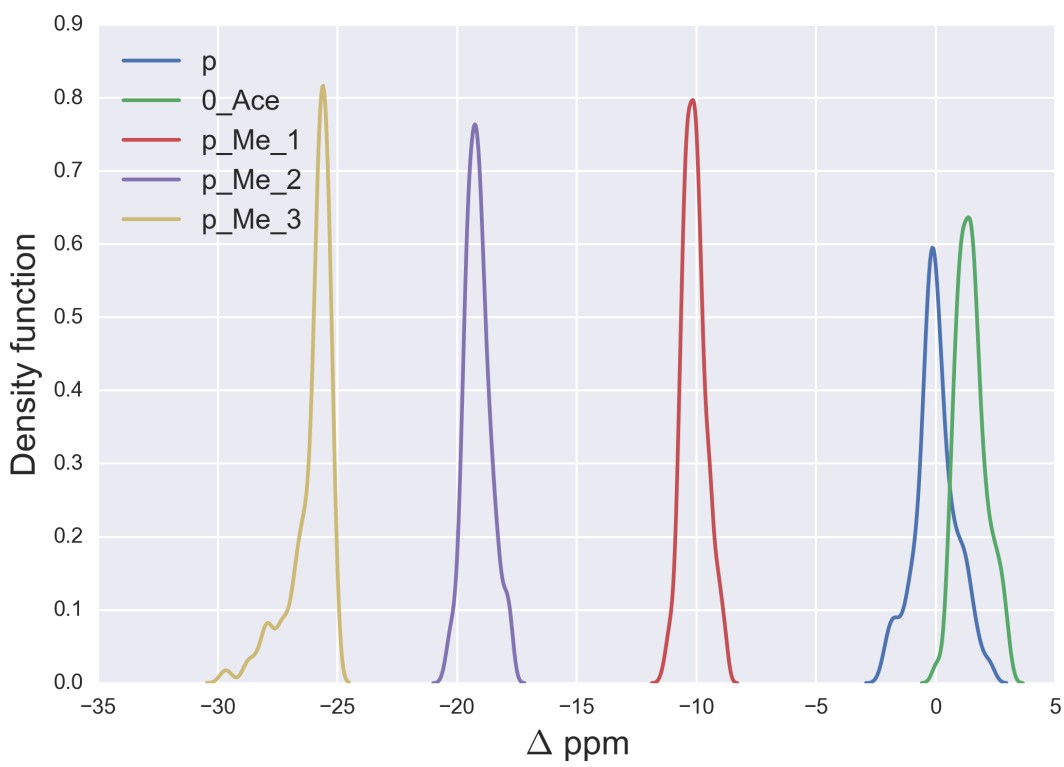

**Figure 1** Kernel Density Estimation of the computed Δ values for the $^{13}C^\varepsilon$ nucleus of *non*-modified charged (blue-line), acetylated (green-line), mono- (red-line), di- (violet-line), and tri-methylated (yellow-line) Lys.

in protein structures (*Martin et al., 2013*); (ii) use as constraints during an NMR-based protein structure determination (*Vila et al., 2008*; *Rosato & Billeter, 2015*); (iii) resolve local inconsistencies between X-ray crystal structures (*Vila et al., 2012*); (iv) determine the tautomer preference of histidine in proteins accurately (*Sudmeier et al., 2003*; *Vila et al., 2011*); (v) study sparsely populated, short-lived, protein states that could play a significant role in protein function (*Hansen & Kay, 2014*); etc.

## MATERIAL AND METHODS

### Preparation of the model tripeptides for the DFT calculations

DFT calculations were carried out for model tripeptides of the form Ace-Gly-**Yyy**-Gly-Nme, with **Yyy** being lysine (Lys) or arginine (Arg). The backbone torsional angles for the tripeptides (including the *N*- and *C*-terminal groups) and the side-chain for residue **Yyy** were taken from a data-base of a non-redundant set of 6,134 high-quality X-ray structures of proteins solved at resolution ≤1.8 Å, with *R* factor ≤0.25, and with less than 30% sequence identity. This ensures that the model tripeptides are a representative sample of the torsional angles observed in nature for the given amino acids. To test that 500 conformations are indeed representative of the conformational accessible-space, we performed a preliminary test for 500 and 1,000 conformations confirming that using 500 rather than 1,000 conformations leads to the same distributions of shielding values but

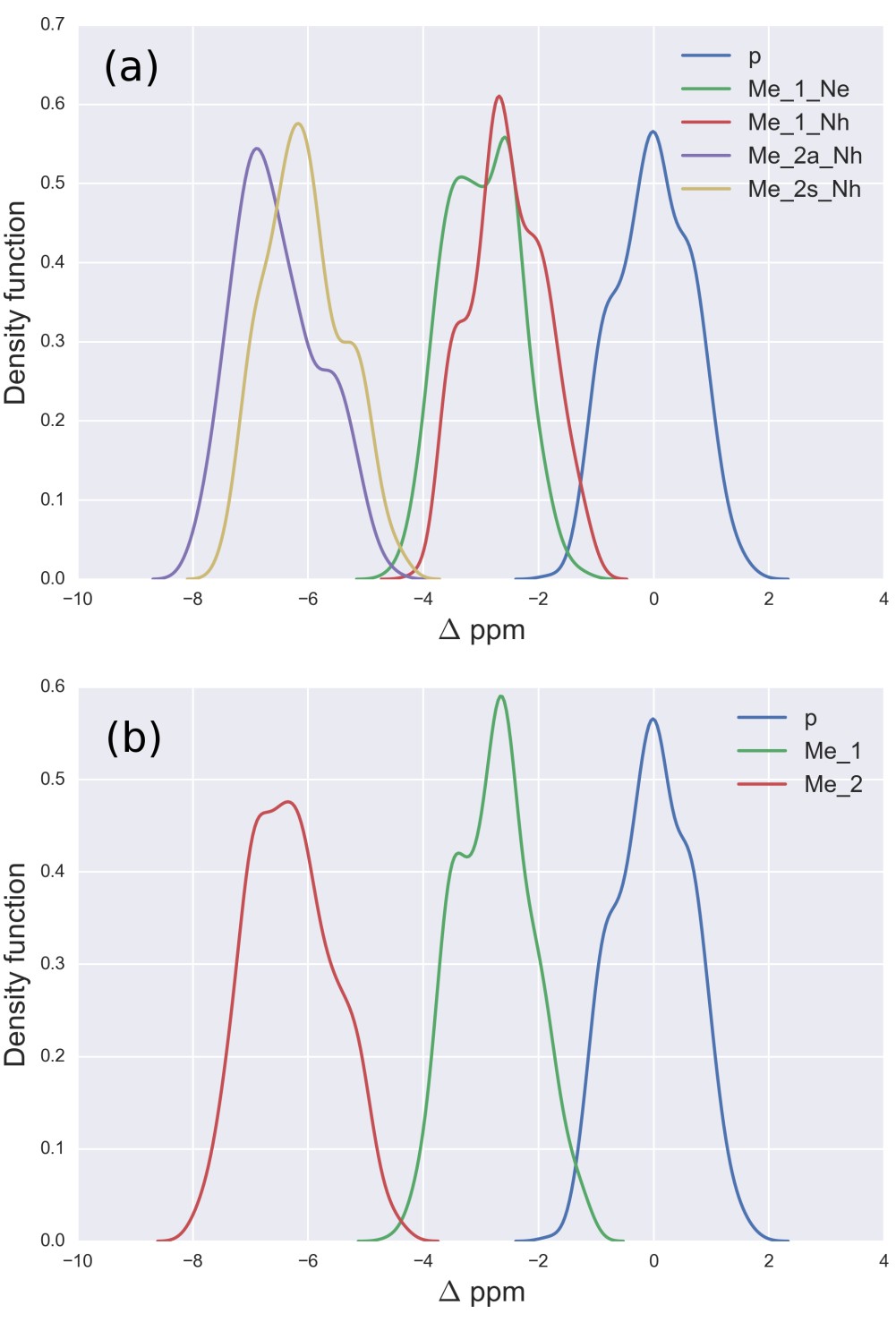

**Figure 2** (A) Kernel Density Estimation of the computed Δ values for the $^{13}C^\zeta$ nucleus of *non*-methylated charged (blue-line), mono-methylated ($N^\varepsilon$ (green-line) and $N^\eta$ (red-line)) and *di*-methylated (symmetric (yellow-line) and asymmetric (violet-line)) Arg; (B) all five curves shown in (A) are condensed in three curves.

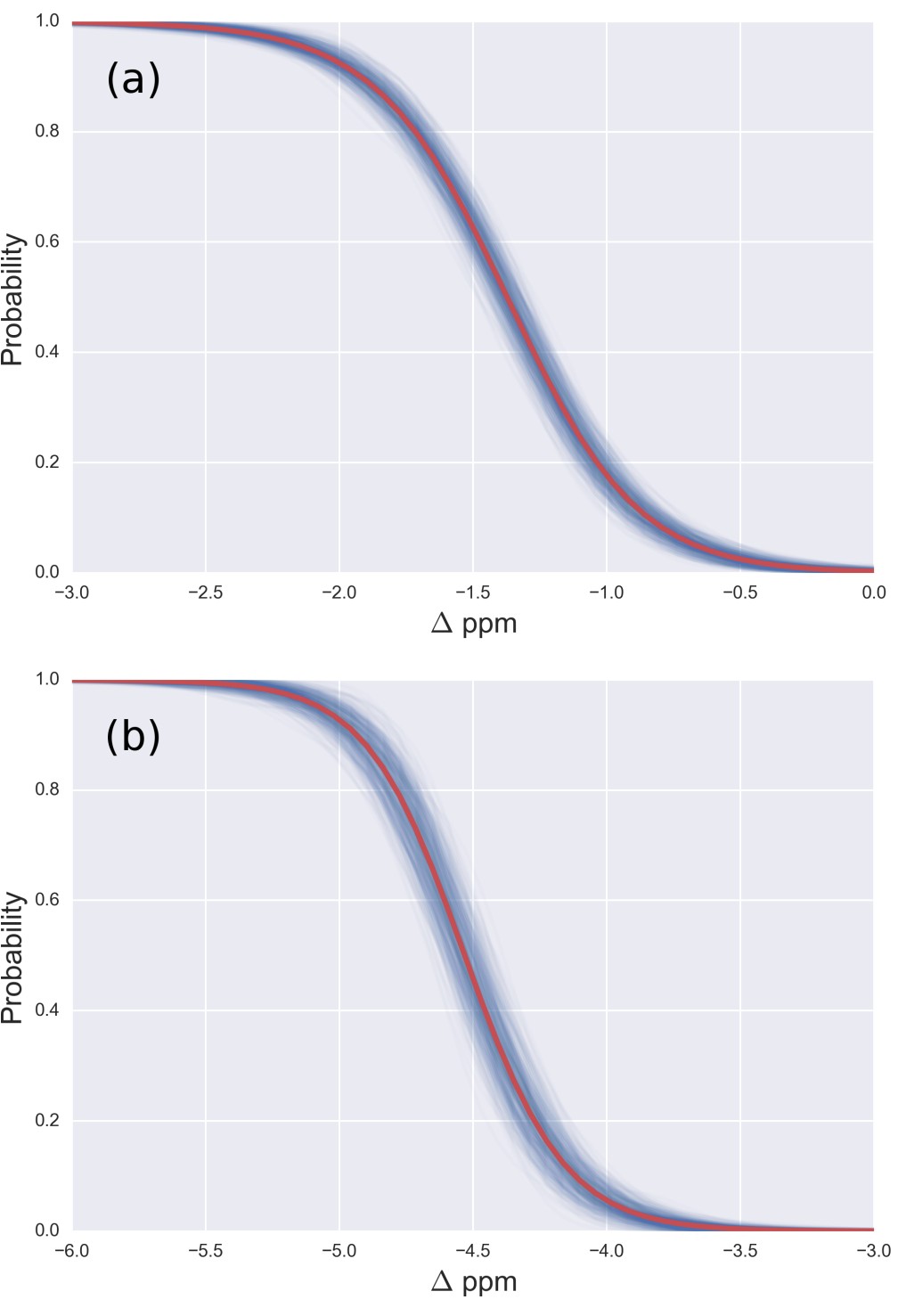

**Figure 3** (A) Probability profile of the Arg residue to be *mono*-methylated (instead of being non-modified) as function of the Δ values for the $^{13}C^{\zeta}$ nucleus; with data from **Fig. 2B**; (B) same as (A) for the *di*-methylated Arg. The red line represents the expected probability-profile and the blue lines the uncertainty in the data according to the Bayesian model.

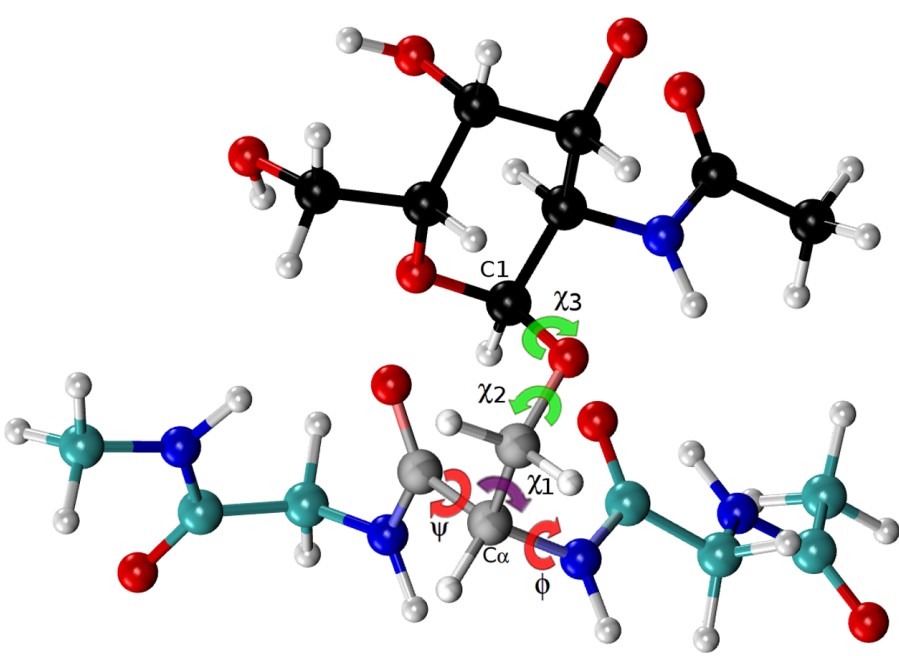

**Figure 4** **Ball and stick representation of a glycan-amino acidic residue, namely for $\alpha$-D-GalpNAc-(1-O)-Ser with "1" representing C1 of the glycan, and "O" representing the oxygen of the side-chain of Ser in an Ac-Gly-Ser-Gly-Nme tripeptide, in an arbitrary conformation.** The $\chi 2$ and $\chi 3$ torsional angles, of the carbohydrate group ($\alpha$-D-GalpNAc), are highlighted in green, while the corresponding one for the amino-acidic residue (Ser) are in red, for $\phi, \psi$, and purple, for $\chi 1$.

with a considerable reduction in computational time. All the 500 conformations were free of atomic-overlaps. It is worth noting that, during the generation of the conformations, the most frequently observed rotamers were used, namely two for acetylation and up to three for methylation. Although the environment may play a role setting preferences for some rotamers this effect cannot be taken into account in a general theoretical analysis because this would imply that every possible environment be taken into account. Because of this, we considered each rotamer as equally probable.

For model Lys tripeptides, we generated a total of 5,000 conformations, namely 500 for charged Lys (i.e., the unmodified amino acid), 1,000 for acetylated Lys, i.e., 500 for each of the rotamers, namely, 0° and 180°, 1,500 conformations for *mono* methylated Lys, i.e., 500 for each of the rotamers, namely, +60°, −60°, and 180°, 1,500 conformations for *di* methylated Lys, i.e., 500 for each of the rotamers, namely, +60°, −60°, and 180° *and* finally 500 for *tri* methylated Lys. The following is the reason for the need to compute more than 500 conformations for modified residues, except for tri-methylated Lys. The replacement of hydrogens by methyl or acetyl groups introduces an asymmetry in the molecule that could influence the DFT computations; hence, rotamers must be generated and the DFT-computed shieldings have to be averaged over these rotameric states.

For Arg, we analyzed a total of 6,000 conformations, namely 500 for charged Arg (i.e., the unmodified amino acid), 2,500 conformations for *mono*-methylated Arg (*Zobel-Thropp, Gary & Clarke, 1998*; *Bedford & Clarke, 2009*), i.e., 1,000 for the *mono*-methylation of each $N^{\eta}$ of the guanidine nitrogens and 500 for the methylation of the $N^{\varepsilon}$ side-chain nitrogen,

respectively, and 3,000 for *di*-methylated Arg, i.e., 2,000 for asymmetric and 1,000 for symmetric *di*-methylation of Arg, respectively.

It is worth noting that, because we are interested *only* in the chemical-shift differences ($\Delta$), the implicit assumptions, during the quantum-chemical calculation of the shieldings, are that most of the errors associated with issues not-included in the calculations, such as those derived from a suitable selection of (i) dielectric solvent; (ii) geometry optimization; (iii) reference value, etc., should not affect the accuracy of the calculations of interest because *all* these effects are expected to be canceled-out during the computation of $\Delta$. However, some intermolecular interactions not-included in the calculations, such as those with explicit solvent molecules and amino acids close in space, may affect the DFT-computed chemical shifts in a way that cannot be canceled out by computing $\Delta$. Nevertheless, the good agreement of the predictions with the observed values for methylation and acetylation of Lysine and Arginine, and *O*-glycosylation of Thr, suggest that oversight of such interactions may not be crucial.

## Preparation of the glyco-amino acidic residue for the DFT calculations

From *all* possible tripeptides of the above mentioned data-base, i.e., of the *non*-redundant set of 6,134 high-quality X-ray structures of proteins, we randomly selected those containing serine (Ser), threonine (Thr) or asparagine (Asn) as residue **Yyy** in the sequence Ace-Xxx-**Yyy**-Zzz-Nme, with Xxx and Zzz being the nearest-neighbor residues of **Yyy** in the selected tripeptide. The backbone torsional angles for the tripeptides (including the *N*- and *C*-terminal groups) and the side-chain for the residue **Yyy** were taken from the above mentioned data-base, while side-chain torsional angles for the residues Xxx and Zzz, that show non atomic-overlaps, were arbitrarily chosen. This procedure ensures that the model tripeptides are a representative sample of the torsional angles observed in nature for a given **Yyy** amino acid. At this point, it is worth noting that, for Lys and Arg, the analysis was carried out on selected tripeptides with the sequence Ace-Gly-**Yyy**-Gly-Nme rather than on Ace-Xxx-**Yyy**-Zzz-Nme tripeptide, as for the glycosylated residues. The reason is that methyl and acetyl groups are small chemical groups while glycans are very bulky moieties and, hence, the degree of freedom of the glycosylated residue (**Yyy**) will be severely restricted depending on the identity of the nearest-neighbor Xxx and Zzz residues. Another peculiarity of the generation of model tripeptides for glycosylated residues is that, after glycosylation of the residue **Yyy**, new side-chain rotations must be explored because of the appearance of additional torsional angles, namely $\chi2$, $\chi3$ for Ser and Thr and $\chi4$ for Asn (see Fig. 4, and Figs. S6 and S7 of the Supplemental Information). We explore these angles at 3 positions, $+60°$, $-60°$, and $180°$. Among all possible conformations only 500, showing non atomic-overlaps, were considered for the computation of the shieldings at the DFT-level of theory. To assure that the computed shielding differences ($\Delta$) mirror *only* the presence of a glycan linked to a **Yyy** (Ser/Thr/Asn) amino acid residue, the monosaccharide of each of the 500 chosen glycosylated conformations was removed and, for the remaining *non*-glycosylated residue, the shieldings were computed at the DFT-level of theory by using the same basis set and functional as for the glycosylated-residue. In

this way, we have generated an ensemble of 500 conformations of glycosylated and 500 conformations of *non*-glycosylated residues, namely for Ser, Thr and Asn, that contain no atomic-overlapping and possessing identical backbone and side-chain torsional angles between the glycosylated and the non-glycosylated residue.

## Computation of the shieldings, for the nuclei of interest, at the DFT level of theory

To compute the gas-phase $^{13}$C-shielding values, at the DFT-level of theory, for any nucleus of interest we will follow the same approach used previously for proteins (*Vila et al., 2009*) and disaccharides (*Garay et al., 2014*), namely, the $^{13}$C shielding value was computed, by using the Gaussian 09 package (Gaussian, Inc, Wallingford, CT) by treating each nucleus, and their neighbors of interest, at the OB98/6-311 + G(2d,p) level of theory, while the remaining nuclei in the sequence were treated at the OB98/3-21G level of theory (*Vila et al., 2009*; *Garay et al., 2014*), i.e., by using the *locally-dense basis set* approach (*Chesnut & Moore, 1989*).

## Computation of the standard deviation from the BMRB

On November 13, 2015, we downloaded *all* the chemical shifts deposited at the Biological Magnetic Resonance Bank (BMRB) (*Ulrich et al., 2008*). We restricted the analysis to entries that were referenced to DSS, TMS or TSP. Then, we re-referenced the chemical shifts to DSS by adding 0.12 ppm to TSP and $-1.7$ ppm to TMS. All data points below $Q1 - 1.5 \times IQR$ or above $Q3 + 1.5 \times IQR$ were considered outliers and removed (with IQR being the Inter-Quartile Range between Q1 and Q3, where Q1 and Q3 are the first and third quartiles, respectively). After removing the outliers, we computed the mean and standard deviation of the distribution for each residue of interest.

## Computation of the probability profiles

During the computation of the probability profile, we assume that each chemical shift belongs to one of two possible Gaussian distributions, for example, due to methylated and not methylated arginine or lysine, respectively. Our aim is to compute a *probability profile* that indicates the probability of a chemical shift to belong to either distribution. For this purpose, we created a simple Bayesian model. In this model, we estimated the mean and standard deviation of the distribution of the chemical-shift differences ($\Delta$), assuming that the $\Delta$ values are distributed approximately as Gaussian distributions with unknown mean and standard deviation.

The *prior* for the mean is a student-*t* distribution with mean equal to the mean of the computed $\Delta$ values and a *scale* equal to 0.35. We assumed this *scale* of 0.35 from the lysine analysis showing that the theoretical and experimental values are in very close agreement within ~1 ppm (see 'Validation Test on Lysine Derivatives'). In other words, we are confident that the theoretical chemical-shift distributions are an accurate representation of the experimental ones within ~1 ppm. Finally, the degrees of freedom of the student-*t* distribution were estimated from the computed $\Delta$ values using as hyper-prior an exponential distribution with mean and standard deviation of 30. A student-*t* distribution with degree of freedom about 30 or larger is almost indistinguishable from a Gaussian

distribution. The *prior* for the standard deviation is a Gamma distribution with mean and standard deviation computed from the experimental values deposited in the BMRB, as explained in the section *Computation of the standard deviation from the BMRB*.

From the model described above, we computed the *posterior* distribution, and from the *posterior* distribution we computed the *posterior* predicted values, i.e., the values of chemical shifts, for each of the two given states, according to the Bayesian model. Given the *posterior* predicted values, it is straightforward to compute the probability of a residue to be in a given state as a function of the $\Delta$ values, essentially because we are assuming only two possible states, i.e., methylated and non-methylated. Figure 3 (and Figs. S3–S5 of the Supplemental Information) shows the results of the analysis in red and blue, semitransparent, lines. Each of these blue lines corresponds to a possible occurrence of the probability profile and the red line, in each of these figures, is the mean of *all* the blues lines. Thus, the red line represents the expected probability profile and the blue lines the uncertainty in the data according to the Bayesian model.

## Data analysis and visualization

Data analysis and visualization were performed using Python (*Van Rossum, 1995*), IPython (*Perez & Granger, 2007*), NumPy (*Van der Walt, Colbert & Varoquaux, 2011*), Pandas (*McKinney, 2010*), Matplotlib (*Hunter, 2007*), and Seaborn (*Waskom et al., 2016*); Bayesian computations were carried out with PyMC3 (*Salvatier, Wiecki & Fonnesbeck, 2016*).

## RESULTS AND DISCUSSION

### Validation test on lysine derivatives

As a first step, it is necessary to validate the methodology. For this purpose, we started by analyzing the computed $\Delta$ values for the $^{13}C^{\varepsilon}$ chemical-shifts of Lys in a model tripeptide, Ace-Gly-Lys-Gly-Nme, for a total of 6,500 conformations of Lys with various degrees of acetylation or methylation (see details of the generation of the conformations in the 'Material and Methods'). By following this procedure, the resulting mean $\Delta$ values from the Kernel Density Estimation of the chemical-shift differences, shown in Fig. 1, are 1.5 ppm, $-10.1$ ppm, $-19.1$ ppm and $-25.8$ ppm for acetylated, *mono-*, *di-* and *tri-*methylated Lys, respectively. A comparison of these computed mean $\Delta$ values with the observed $^{13}C^{\varepsilon}$ chemical-shift variations of charged Lys upon acetylation and methylation, namely, 0.0, 9.0, 18.0 and 26.5 ppm, respectively (*Theillet et al., 2012a*), enables us to conclude that very good agreement exists within $\sim$1 ppm between these theoretical predictions and experimental evidence. Overall, the $^{13}C^{\varepsilon}$ chemical-shifts are sensitive enough to detect methylation (see yellow-, violet-, red- and blue-lines in Fig. 1) but not acetylation states of Lys; the superposition of the $\Delta$ values (see green- and blue-line in Fig. 1) make the distinction between acetylated and *non*-modified Lys unfeasible. At this point is worth noting, from Fig. 1, that the effect of chemical shift differences due to rotamer changes ($\sim$1.3 ppm, on average) is by far smaller than the effect due to PTM changes ($\sim$10 ppm, on average).

In addition, the computed $\Delta$-values for the $^{13}C^{\alpha}$ and $^{13}C^{\beta}$ nuclei upon acetylation and methylation of Lys are shown in Figs. S1A and S1B of Supplemental Information.

The superposition of these curves for the methylated with those of the *non*-methylated charged Lys (Figs. S1A and S1B) indicates that these nuclei are not sensitive enough to detect methylation. From Figs. S1A and S1B (of Supplemental Information) we also observe that the curves for acetylated Lys do not fully-overlap either the ones for methylated or the *non*-modified charged Lys and, hence, the origin of this behavior must be investigated. In this regard, it should be noted that acetylation, but not methylation, does not preserve the state of charge of Lys. Consequently, the change in protonation upon acetylation should be the reason for the above unexpected result. Indeed, the change of protonation for *non*-modified Lys, as occurs at a high pH value, leads to a $\Delta$ distribution (see Figs. S1C and S1D) showing a very similar pattern to the one obtained after acetylation (see Figs. S1A and S1B). Taking all this together, these results indicate that the $^{13}C^{\alpha}$ and $^{13}C^{\beta}$ nuclei of Lys are not sensitive enough to detect either methylation or acetylation states of lysine in proteins.

Having presented the above test on lysine, we have a validation of our theoretical approach and have shown that computation of the $\Delta$ values, for a given nucleus, is a useful method with which to detect the methylation states of Lys (*Theillet et al., 2012a*; *Theillet et al., 2012b*), but not acetylation (*Theillet et al., 2012a*). Consequently, we decided to extend this analysis to discuss methylation of Arg, and glycosylation of Ser, Asn and Thr, with model tripeptides, and the results are discussed below.

## Methylation of arginine

For Arg, with the sequence Ace-Gly-Arg-Gly-Nme, we have computed the chemical-shifts for the $^{13}C^{\zeta}$ nucleus in 6,000 conformations (see details of the generation of the conformations in 'Materials and Methods'). The Kernel Density Estimation (KDE) of the $\Delta$ are shown in Fig. 2A. This figure shows that it is not possible to distinguish between *mono*-methylated, i.e., between Arg methylated at the $N^{\varepsilon}$ or $N^{\eta}$ group, respectively, or *di*-methylated, i.e., between symmetric or asymmetric *di*-methylated Arg. As a consequence, *all* 5 curves shown in Fig. 2A can be condensed into 3 curves, shown in Fig. 2B. Each of the resulting 3 curves shows the $\Delta$ values for the *non*-, *mono*-, and *di*-methylated Arg, respectively. A comparison among the resulting distributions enables us to infer that *non*-, *mono*- and *di*-methylated Arg can be distinguished by monitoring the chemical-shift variations of the $^{13}C^{\zeta}$ nucleus. However, as shown in Fig. 2B, there is still a small overlap between the distributions and, hence, regions of ambiguity. Because of this overlapping, we compute a probability profile i.e., the probability that Arg is in one of two possible states as a function of its $\Delta$ value. Computation of the Arg probability-profile is illustrated by a red line and their uncertainty as blue lines (see Fig. 3). In general, the blue lines represent different occurrences of the probability-profiles, and the red line the average over all of them (see Figs. 3 and S3–S5 of the Supplemental Information). Thus, chemical-shift differences ($\Delta$) within the range −2 ppm to −3 ppm indicate a large probability (>80%) that Arg is *mono*-methylated (see red-line of Fig. 3A) and a very low probability (∼0%) of being *di*-methylated (see Fig. 2B). On the other hand, $\Delta$ values smaller than −4.8 ppm indicate a large probability (>80%) that Arg is *di*-methylated (see red-line of Fig. 3B) and very low probability of being *mono*-methylated (see Fig. 2B). To proceed further with Arg analysis
we find that the $\Delta$ values for the $^{13}C^{\alpha}$ and $^{13}C^{\beta}$ nuclei upon Arg methylation (shown in Figs. S2A and S2B of Supplemental Information) are superimposed among themselves indicating, as for Lys, that none of these nuclei is sensitive enough to detect methylation.

At this point, it is worth noting the following. First, there is no significant $pK_a$ change within $\sim$0.5 pK units among *mono*-methylated, *di*-methylated (symmetric or asymmetric) *and non*-methylated Arg (*Evich et al., 2015*). Therefore, perturbation of the $pK_a$ upon methylation is not large enough to be used as a probe with which to sense Arg modification. Second, there are other nuclei, than carbons, of the Arg side-chain, such as $N^{\varepsilon}$, that show large chemical-shift dispersion upon methylation (*Theillet et al., 2012b*). However, as noted by *Theillet et al. (2012b)*, there is some limitation in using this nucleus to detect methylation: "…*NMR detection of solvent accessible protein arginine* NH$\varepsilon$ *and* NH$\eta$ *resonances is only feasible at pH lower than 6.5, because of fast water/guanidinium proton chemical exchange*…". This drawback prevents the use of these nuclei to sense PTMs in proteins around physiological conditions, where most of the biological activities take place and which are conditions desirable for many experiments such as arginine methylase activity measurements. Despite the limitation cited by *Theillet et al. (2012b)*, in uniformly $^{13}C$- and $^{15}N$-labeled proteins, the $^{15}N$ chemical shifts can also be obtained from experiments that do not rely on the exchangeable, and therefore, pH-sensitive protons, e.g., from HCN-type experiments (*Fiala, Czernek & Sklenář, 2000*).

To end, the computation of the probability profiles was carried out taking into account the chemical shifts of *only* two states, such as *mono*- or *di*-methylation, but not other possible modifications, such as phosphorylation or citrullination, which were not considered in our analysis.

## Glycosylation of Ser, Thr and Asn

Finally, we explore whether the computed $\Delta$ values upon-glycosylation, at the DFT-level of theory (*Garay et al., 2014*), for the $^{13}C$ nucleus closer to the glycosylation site, namely $^{13}C^{\beta}$ for the Ser and Thr and $^{13}C^{\gamma}$ for the Asn residue, respectively, can be used as a probe with which to sense the most commonly seen *O*- and *N*-glycosylation, namely the *O*-linked *N*-acetylglucosamine (GlcpNAc) and *N*-acetylgalactosamine (GalpNAc) glycosylation of Ser and Thr (*Nishikawa et al., 2010*), and the *N*-acetylglucosamine glycosylation of Asn (*Chauhan, Rao & Raghava, 2013*). By focusing our attention on the $\Delta$ values upon glycosylation for some selected $^{13}C$ nuclei of the residue side-chain, we will be able to determine whether, first, the $\Delta$-values can be used to determine glycosylation and, second, the type of glycosylated residue, e.g., GlcpNAc or GalpNAc for Ser and Thr. By focusing our analysis on some nuclei of the amino-acid residue side-chain, rather than on the $^{13}C$ nuclei of the monosaccharide, to which the residue is linked, would avoid comparing the computed $^{13}C$ chemical shifts of residue-linked glycans with those from *non*-linked glycans for which, as far we know, there is very sparse information.

## O-glycosylation of Ser

We started the glycosylation analysis by computing the $^{13}C$ chemical-shift for an ensemble of 500 conformations, generated as a function of the torsional angles $\phi$, $\psi$, $\chi 1$, $\chi 2$ and $\chi 3$
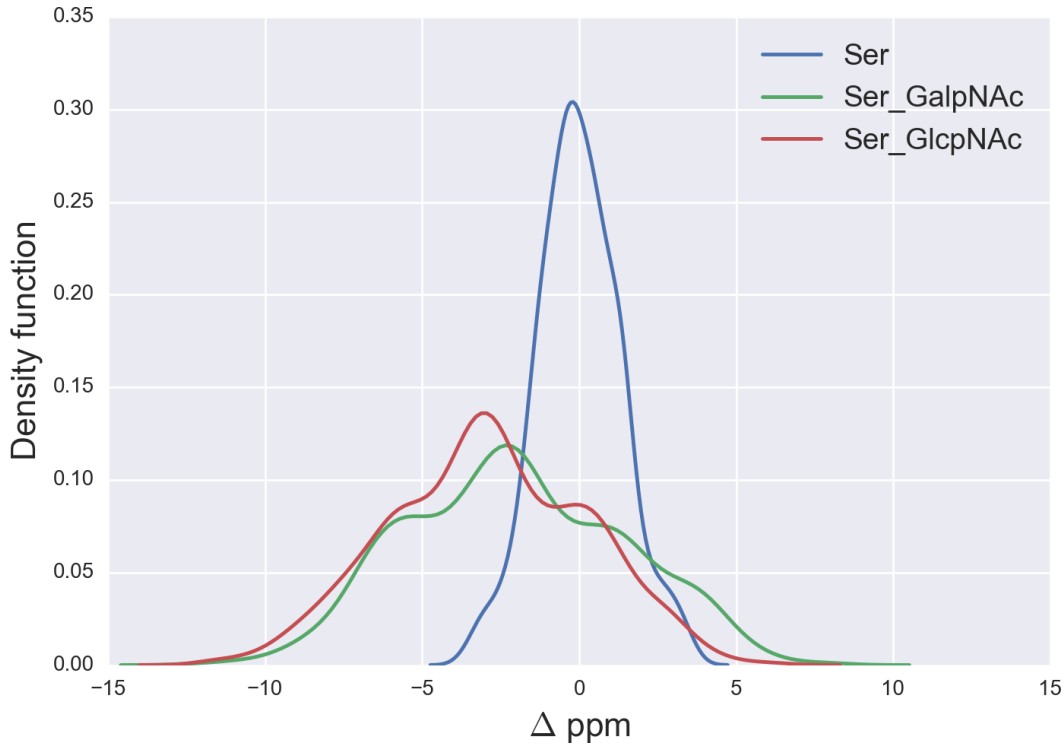

**Figure 5** Kernel Density Estimation of the computed Δ values for the $^{13}C^{\beta}$ nucleus of Ser for: Ace-Xxx-Ser-Zzz-NMe (blue-line), $\alpha$-D-GalpNAc-(1-O)-Ser (green-line) and $\beta$-D-GlcpNAc-(1-O)-Ser (red-line).

(see Fig. 4), at the DFT-level of theory, for both the glycosylated and the non-glycosylated Ser. Then, the $^{13}C$ chemical-shift differences, Δ, for Ser were computed, i.e., between the $^{13}C$ chemical shift for the non-glycosylated Ser in the tripeptide Ac-Xxx-Ser-Zzz-Nme, with Xxx and Zzz being the nearest-neighbor residues of Ser in the selected tripeptide, from a non-redundant set of high-quality 6,134 X-ray structures of proteins, and the corresponding $^{13}C$ chemical shift for glycosylated Ser, namely for $\alpha$-D-GalpNAc-(1-O)-Ser and $\beta$-D-GlcpNAc-(1-O)-Ser, with Ser in the Ac-Xxx-Ser-Zzz-Nme tripeptide. The identical procedure, to that of Ser, was also carried out for 500 conformations of both the isolated Thr (Ac-Xxx-Thr-Zzz-Nme) and the glycosylated Thr, namely $\alpha$-D-GalpNAc-(1-O)-Thr and $\beta$-D-GlcpNAc-(1-O)-Thr, with Thr in the Ac-Xxx-Thr-Zzz-Nme tripeptide, and 500 conformations of both the isolated Asn (Ac-Xxx-Asn-Zzz-Nme) and the glycosylated Asn, namely for $\beta$-D-GlcpNAc-(1-N)-Asn, with Asn in the Ac-Xxx-Asn-Zzz-Nme tripeptide. The resulting curves for the Δ values are shown in Figs. 5 and 6 for the $^{13}C^{\beta}$ of Ser and Thr, respectively and Fig. 7 for the $^{13}C^{\gamma}$ of Asn.

From Fig. 5 we can see, first, large overlapping Δ values for glycosylated Ser, namely between the $\alpha$-D-GalpNAc-(1-O)-Ser and the $\beta$-D-GlcpNAc-(1-O)-Ser (shown as green- and red-lines, respectively, in Fig. 5) and, second, a broad distribution of the Δ values for glycosylated Ser with respect to non-glycosylated Ser (blue-line in Fig. 5). The large overlapping of Δ values between the glycosylated curves for Ser enables us to represent both

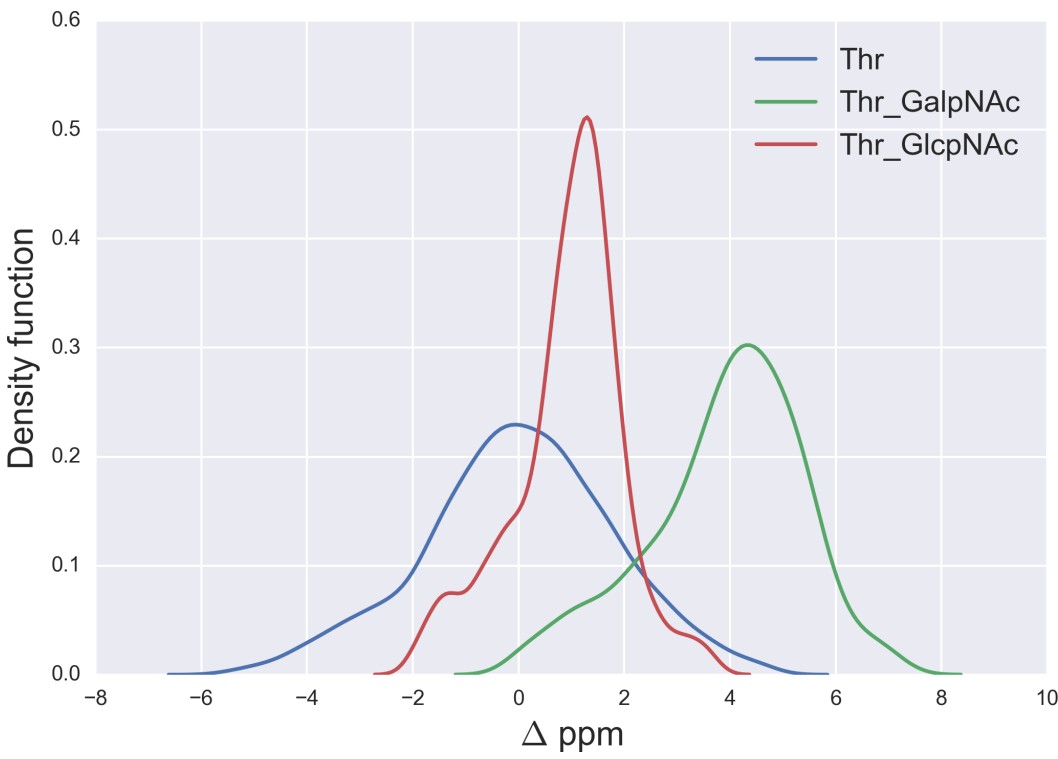

**Figure 6** Kernel Density Estimation of the computed Δ values for the $^{13}C^{\beta}$ nucleus of Thr for: Ace-Xxx-Thr-Zzz-NMe (blue-line), $\alpha$-D-GalpNAc-(1-O)-Thr (green-line) and $\beta$-D-GlcpNAc-(1-O)-Thr (red-line).

kinds of glycosylation as a single curve and, hence, a unique distribution of glycosylation probability (see Fig. S3 of Supplemental Information). As a result, a Δ value smaller than −3 ppm indicates a large probability (>80%) that Ser is glycosylated (see red-line in Fig. S3 of Supplemental Information). However, for Δ values above 2 ppm the uncertainty in the probability of glycosylation (represented by the blue-lines in Fig. S3 of Supplemental Information) grows, thus preventing us from making an accurate assessment as to whether Ser is glycosylated. This is a consequence of the overlapping Δ values between the $\alpha$-D-GalpNAc-(1-O)-Ser and *non*-glycosylated Ser (see Fig. 5).

### O-glycosylation of Thr

A similar analysis for the Δ values of the $^{13}C^{\beta}$ of Thr, shown in Fig. 6, indicates that *N*-acetylgalactosamine glycosylation of Thr can be detected, mainly because there is no strong overlapping between the glycosylated [$\alpha$-D-GalpNAc-(1-O)-Thr] and the non-glycosylated (Ace-Xxx-Thr-Zzz-NMe) Δ-distribution for Thr. Indeed, if the computed Δ value is larger than ∼+3 ppm there is >80% probability that an *N*-acetylgalactosamine glycosylation of Thr exists (see Fig. S4 of Supplemental Information). On the other hand, detection of *N*-acetylglucosamine glycosylation of Thr is not straightforward because of the strong overlapping of the Δ distributions between the $\alpha$-D-GlcpNAc-(1-O)-Thr and the non-glycosylated Thr (see Fig. 6).

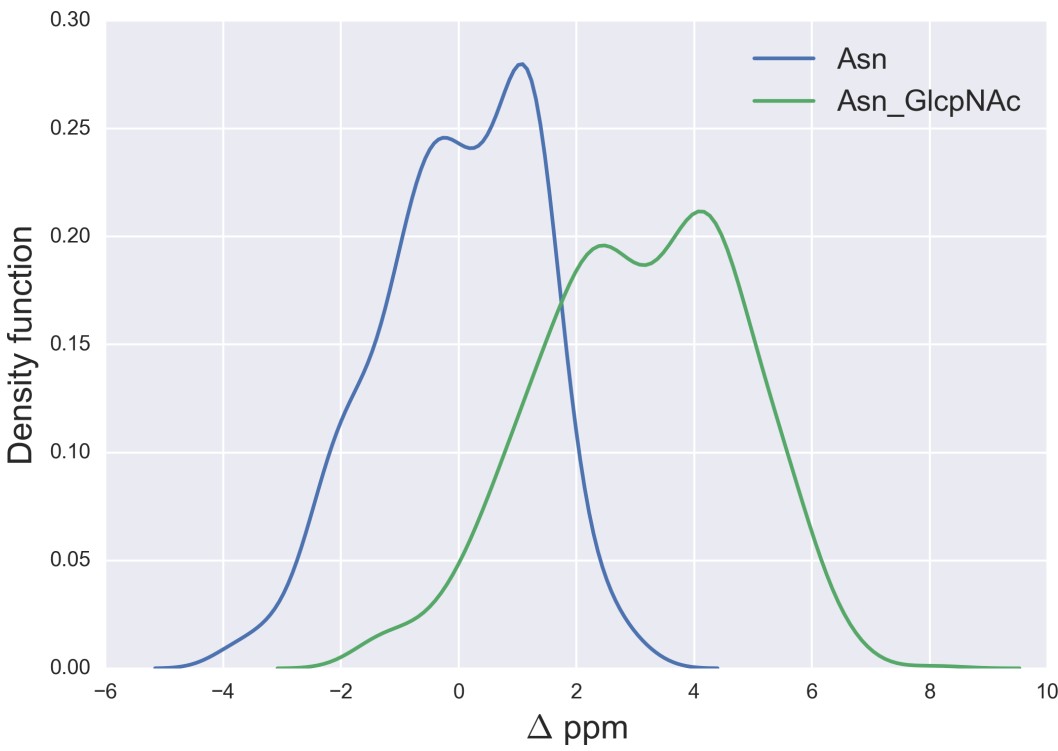

**Figure 7** Kernel Density Estimation of the computed Δ values for the $^{13}C^{\gamma}$ nucleus Asn for: Ace-Xxx-Asn-Zzz-NMe (blue-line) and $\beta$-D-GlcpNAc-(1-N)-Asn (green-line).

The large $^{13}C^{\beta}$ chemical shift difference observed for Thr-106 upon glycosylation ($\Delta = +9.9$ ppm), in the GalNAc$\alpha$-IFN$\alpha$2a glycoprotein (*Ghasriani et al., 2013*), is fully consistent with our prediction for the *N*-acetylgalactosamine glycosylation of Thr. Indeed, a $\Delta > 6$ ppm (see Fig. S4 of Supplemental Information) reveals a high probability for Thr being glycosylated. However, it should be noted that $\Delta$'s $> 8$ ppm are missing from Fig. 6, e.g., as for Thr-106 ($\Delta \sim 10$ ppm). At this point, there are two problems associated with the analysis of Thr-106 glycosylation that needs to be clarified, namely the meaning of the computed $\Delta$ distributions and whether the observed $\Delta$ value for Thr-106 can be reproduced by our calculations. Let us address each of them separately. First, the $\Delta$ distributions in Fig. 6, like any other distributions inferred in this work (see Figs. 1, 2 and 5–7; S1 and S2 of Supplemental Information) are meant to be representative of the chemical shift population for modified and unmodified residues, respectively, and, hence, $\Delta$ values out of range in these figures need to be interpreted as events with low probability, rather than null, occurrence. Second, to test whether the observed $\Delta$ value for Thr-106 can be reproduced we decided to (i) compute the chemical-shift values for the tripeptide Ac-Val-Thr$_{106}$-Glu-Nme, with Val and Glu being the nearest-neighbor amino-acid residues in the nonglycosylated and glycosylated GalNAc$\alpha$-IFN$\alpha$2a protein sequence; and (ii) adopt, for the tripeptide, the torsional angles defined for each of the 20 nonglycosylated conformations (PDB id 1ITF) and the 24 glycosylated conformations (PDB id 2MLS) of the protein. As a result, we obtain for Thr-106 a

computed averaged $\Delta$ value ($\sim$12 ppm) in close agreement, within $\sim$2 ppm, with the observed one ($\sim$10 ppm). At this point, is worth noting that the standard deviation (*sd*) of the computed chemical-shift**s** for the nonglycosylated conformations is quite large ($\sim$3 ppm); in fact, this is significantly larger than that the *sd* computed for the glycosylated conformations ($\sim$1 ppm) and, hence, consistent with the observation that nonglycosylated conformations are more flexible than that the glycosylated one (*Ghasriani et al., 2013*).

A comparison of the *N*-acetylgalactosamine and *N*-acetylglucosamine glycosylation of Ser and Thr (see red and green lines in Figs. 5 and 6, respectively) highlight two very different behaviors, in terms of $\Delta$, albeit Ser and Thr side-chains differ *only* by the attached chemical-group to the $C^\beta$ nucleus, namely an H and a $CH_3$ group, respectively (see Figs. 4 and S6 of Supplemental Information). Actually, this fact can be understood in light of the differences, in term of the side-chain accessible conformational space, between glycosylated Ser and Thr. Indeed, the 500 conformations of glycosylated Ser possess the side-chain $\chi 2$ torsional-angle equally clustered among $-60°$, $+60°$ and $180°$, respectively, independent of the nature of the attached glycan. On the contrary, the $\chi 2$ torsional-angles of the 500 conformations of either $\alpha$-D-GalpNAc-(1-O)-Thr or $\beta$-D-GlcpNAc-(1-O)-Thr are mostly clustered around $+60°$ or $180°$, respectively.

### N-glycosylation of Asn

Finally, the $\Delta$ values for the $^{13}C^\gamma$ of Asn are shown in Fig. 7. There is no full overlapping between $\Delta$ values computed from glycosylated and non-glycosylated Asn; hence, a chemical-shift difference larger than $\sim$2 ppm indicates a large probability ($>$80%) of Asn being glycosylated (see Fig. S5 of Supplemental Information).

## CONCLUSIONS

The prediction of methylation and acetylation of lysine is in good agreement (within $\sim$1 ppm) with the NMR-observed values (*Theillet et al., 2012a*). Moreover, the prediction for the *N*-acetylgalactosamine glycosylation of Thr was found to be consistent with the observed values (within $\sim$2 ppm) for the glycosylation of Thr-106 in the GalNAc$\alpha$-IFN$\alpha$2a glycoprotein (*Ghasriani et al., 2013*).

Monitoring the $\Delta$'s of the: (i) $^{13}C^\zeta$ nucleus of arginine enables us to distinguish *non-*, *mono-* and *di*-methylated Arg; and (ii) $^{13}C^\beta$ nucleus of Ser and Thr and the $^{13}C^\gamma$ nucleus of Asn, can be used to detect the most commonly seen *O*- and *N*-glycosylations of these residues, except for the type of monosaccharide linked to Ser. In addition, to solve the overlapping problem between $\Delta$ curves the probability profiles enable us to estimate the chance that the residue is modified.

The chemical-shift is a local property and, hence, the proposed detection method should be useful for any state of the protein, even for intrinsically disordered proteins (IDP) because, to the best of our knowledge, the conformational distribution of IDP's is the same as that of non IDP proteins. Consequently, our analysis is expected to be valid for both structured and non-structured proteins. This mentioned property of the IDP enables us to hypothesize that detecting PTM of an IDP should be easier than for non-IDP

proteins, and the reason for this assumption follow. Experimentally there are, at least, two contributions to the changes in the chemical shifts, one from the torsional-angle variations and the other one from the PTM. Without accurate knowledge of the protein structure the best way to remove the torsional-angle contribution is to average over the largest possible ensemble of conformations. In this regard, the intrinsic larger conformational averaging of the IDP provided an advantage, over regular structured proteins, to detect PTM.

A major drawback of our approach is that $^{13}$C-labeled eukaryotic proteins are typically expressed as recombinant proteins in bacterial systems, which usually lack the ability to introduce eukaryotic PTMs.

Overall, with a test on lysine derivatives, the strategy proposed here to detect acetylation of Lys, methylation of Lys and Arg, and the *O*- and *N*-glycosylation of Ser, Thr and Asn residues has the potential to be used for recognition of posttranslational modifications within living cells (*Doll et al., 2016*), e.g., by using the proposed $^{13}$C NMR spectroscopic methodology in cells for the study of intrinsically disordered proteins (*Felli, Gonnelli & Pierattelli, 2014*).

## ACKNOWLEDGEMENTS

The authors thank the reviewers for their valuable comments and criticisms.

### Funding

This research was supported by grants from the US National Institutes of Health (GM-14312), the US National Science Foundation (MCB10-19767) (HAS), and PIP-112-2011-0100030 from CONICET-Argentina, Project 3-2212 from UNSL-Argentina, and PICT-2014-0556 from ANPCyT-Argentina (JAV). The funders had no role in study design, data collection and analysis, decision to publish, or preparation of the manuscript.

### Grant Disclosures

The following grant information was disclosed by the authors:
US National Institutes of Health: GM-14312.
US National Science Foundation: MCB10-19767.
CONICET-Argentina: PIP-112-2011-0100030.
UNSL-Argentina: 3-2212.
ANPCyT-Argentina.: PICT-2014-0556.

### Competing Interests

The authors declare there are no competing interests.

### Author Contributions

- Pablo G. Garay performed the experiments, analyzed the data, prepared figures and/or tables, reviewed drafts of the paper.
- Osvaldo A. Martin conceived and designed the experiments, analyzed the data, wrote the paper, prepared figures and/or tables, reviewed drafts of the paper.

- Harold A. Scheraga analyzed the data, wrote the paper, reviewed drafts of the paper.
- Jorge A. Vila conceived and designed the experiments, analyzed the data, wrote the paper, reviewed drafts of the paper.

### Data Availability

GitHub: https://github.com/aloctavodia/PTM.

### Supplemental Information

Supplemental information for this article can be found online at http://dx.doi.org/10.7717/peerj.2253#supplemental-information.

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
