# Peer review of "Detection of methylation, acetylation and glycosylation of protein residues by monitoring 13C chemical-shift changes: A quantum-chemical study"

_PeerJ, doi:10.7717/peerj.2253_

## Round 0.1 · original submission · Major Revisions

Both reviewers gave a very detailed list of comments and suggestions. Please try to address them all and in case you feel that a recommendation would be out of scope of this manuscript, try to argue why you cannot follow the reviewers suggestion. I would like to especially point out the remark of reviewer 1 with respect to the validity of the findings where he comments on your statement that "all errors associated with issues not-included in the calculations". I consider it very important to discuss and address the concerns of the reviewer in this case. I also agree with the remark of this reviewer that the conclusions should be improved.

·

Basic reporting

The authors addressed an important and so far not very exploited issue of the potential of the chemical shifts to indicate post-translational modifications (PTMs). The study does not cover some effects (e.g. solvation), which is understandable and acceptable, but I am not convinced that all these effects are cancelled out as the author claim (see "Validity of the Findings"). In general, I find the presented results useful for the scientific community, but I feel that some parts of the manuscript need to be improved:

(1) The title and abstract do not clearly say what was actually done by the authors. "Detection of methylation, acetylation and glycosylation of protein residues by monitoring 13C chemical-shift changes" sounds like a title of an experimental study. It is not mentioned in the abstract that the authors used QM computations (DFT in particular) to calculate the chemical shifts: "theoretical computations" could be e.g. predictions by programs like SPARTA, SHIFTX etc. The sentence "we compare our theoretical computations of the 13Cepsilon chemical-shift values against experimental data" does not specify that the experimental data were taken from the literature, not obtained by the authors.

(2) Conclusions should be re-written to clearly summarize the findings of the authors, stating which results support the conclusions.

Experimental design

While the authors overall present a very careful analysis of the post-translation modification effects on 13C chemical shifts, I see several points that should be clarified and/or discussed in more detail in the manuscript:

(1)  More conformations are generated for the modified than for the non-modified residues and the reasons for this choice are well explained in the text. However, it's is puzzling that results are then presented as differences = (chemical shift of each conformation of the modified residue)-(average over all conformations of the non-modified residue). As a result, average backbone conformation of a non-modified residue is compared to the immediate conformation of the modified one and the calculated chemical shift differences thus reflect a mix of 2 effects: (1) backbone conformation and (2) post-translation modification. I would prefer to have the two effects separated to clearly see the influence of the acetylation/methylation etc. on the 13C chemical shift only.

(2) Could the authors provide more information (possibly in the Supporting info) on how the rotameric states were generated, e.g. the step size used during the rotation. Were all possible rotamers used? Some rotamers may potentially be preferred over others due to interactions with the surroundings while other rotamers may be only sparsely populated. Are the differences in the chemical shift because of the rotation smaller than effects caused by different types of the post-translation modifications?

Validity of the findings

(1) The authors claim that "all errors associated with issues not-included in the calculations" are eliminated since only differences in the calculated chemical shifts are presented (rather than absolute values). While this is certainly true for referencing and the choice of methodology (DFT functional/basis set combination), I do not agree with the statement especially as far as the effects of the explicit solvent and surroundings are concerned. While the effect of the bulk of solvent and the protein can be eliminated in this way, the interactions with explicit solvent molecules and amino acids close in space can be very different for the modified and non-modified residues. If the authors claim that these different interactions do not change the distributions of chemical shift differences observed for different types of modifications, they should provide a clear evidence (a test computation).

(2) Although I agree the differences in the chemical shifts is most useful, comparison of the absolute values of the calculated vs. experimental chemical shifts should be presented at least for some test cases (e.g. in the Supporting info) to give an idea about the systematic errors.

Additional comments

page 4: 13C-labeled eukaryotic proteins are typically expressed as recombinant proteins in bacterial systems, which usually lack the ability to introduce eukaryotic PTMs. This complication should be mentioned in the text.

page 12: In uniformly 13C,15N-labeled proteins, the 15N chemical shifts can be also obtained from experiments that do not rely on the exchangeable and therefore pH-sensitive protons, e.g. from 2D C-N correlation experiments with a direct 13C detection, of from HCN-type experiments.

page 16: The risk of false negative results (result of distribution overlaps) should be mentioned

page 16: It is proposed in Conclusions that the presented methodology is applicable for intrinsically disordered proteins. What is the effect of conformational averaging on the distributions? How completely the geometries taken from PDB cover the conformational space of the intrinsically disordered proteins?

Typos:

p 2, l 21 : "1N" should be "1H"
p 6, l 5: "o the" should be "of the"
p 7, l 16: "Arginine" should be "arginine"
p 9, l 9: "Test" should be "test"
p 10, l 19: extra comma after "Delta"
p 13, lines 5 and 7: "shifts?"

Reviewer 2 ·

Basic reporting

.

Experimental design

.

Validity of the findings

.

Additional comments

The author report a method to monitor posttranslational modification of proteins by calculating the 13C chemical shifts. I recommend the publication of the manuscript with minor revisions for the following reasons:

1) The authors report to build the tripeptides from a data-base of a non-redundant set of 6,134 high-quality X-ray structures of proteins for the calculations of Lys and Arg modifications, as stated in the section relative to the model building of glyco-amino acidic residue. The authors should clarify how they build the tripeptides models.

2) For what concerns the calculations of glyco-amino acidic residues, the authors considered the general sequence of tripeptide Ace-Xxx-Yyy-Zzz-Nme to take into account the influence of neighbour residues to the Ser/Thr/Asn. This implies that different tripeptides were considered in the chemical shifts calculations but only one Δ value for the Ser/Thr/Asn mdofications. This point should be also clarified.

3) In the manuscript some typos have been found, for example: “between the 13C chemical shifts? for the non-glycosylated”. Please, carefully revise all the text.

---

## Round 0.2 · Minor Revisions

In my opinion this revised manuscript made a major step forward when compared with the initial version, and you did a good job in responding to the questions and concerns from the reviewers and myself. There are only two small minor issues left.

Firstly, in your rebuttal letter you have asked reviewer 1 if he could provide adequate references with examples of experiments correlating 15N and 13C. The reviewer in his comment below provided three references, and I would recommend to incorporate those that you consider most appropriate into the ms.

I also agree with the reviewer that your response on p. 2, line 5 of the rebuttal letter ("the most frequently observed rotamers...") should be part of the main text.

·

Basic reporting

The authors carefully answered all questions and I can recommend the paper for publication in PeerJ. I just suggest the authors to include their response on p. 2, line 5 of the rebuttal letter ("the most frequently observed rotamers...") in the main text.

My response to p. 3, line 12:

Examples of experiments correlating 15N and 13C

with 1H detection:
Fiala et al., J Biomol NMR, 16 (2000) 291

with the direct 13C detection:
Bermel W. et al., Angew Chem Int Ed, 44 (2005) 3089

with direct 15N detection:
Takeuchi K et al., J Biomol NMR, 47 (2010) 271

Experimental design

no comment

Validity of the findings

no comment

---

## Round 0.3 · accepted · Accept

All remaining remarks/concerns of the reviewers and myself have been addressed to my full satisfaction, and no further review is needed. The manuscript can therefore be accepted for publication in PeerJ without further delay.